# Reliability of Tumor Testing Compared to Germline Testing for Detecting BRCA1 and BRCA2 Mutations in Patients with Epithelial Ovarian Cancer

**DOI:** 10.3390/jpm11070593

**Published:** 2021-06-24

**Authors:** Christine Bekos, Christoph Grimm, Marlene Kranawetter, Stephan Polterauer, Felicitas Oberndorfer, Yen Tan, Leonhard Müllauer, Christian F. Singer

**Affiliations:** 1Comprehensive Cancer Center, Gynecologic Cancer Unit, Department of Obstetrics and Gynecology, Division of General Gynecology and Gynecologic Oncology, Medical University of Vienna, A-1090 Vienna, Austria; christine.bekos@meduniwien.ac.at (C.B.); christoph.grimm@meduniwien.ac.at (C.G.); marlene.kranawetter@meduniwien.ac.at (M.K.); stephan.polterauer@meduniwien.ac.at (S.P.); yen.tan@meduniwien.ac.at (Y.T.); christian.singer@meduniwien.ac.at (C.F.S.); 2Department of Pathology, Medical University of Vienna, A-1090 Vienna, Austria; felicitas.oberndorfer@meduniwien.ac.at

**Keywords:** BRCA mutations, tumor testing, germline testing, variants of uncertain significance

## Abstract

Background: BRCA 1/2 mutation status has become one of the most important parameters for treatment decision in patients with epithelial ovarian cancer (EOC). The aim of this study was to compare tumor DNA with blood DNA sequencing to evaluate the reliability of BRCA tumor testing results. Methods: Patients who were treated for EOC between 2003 and 2019 at the Medical University of Vienna and underwent both germline (gBRCA) and tumor (tBRCA) testing for BRCA mutations were identified. We calculated the concordance rate and further analyzed discordant cases. Results: Out of 140 patients with EOC, gBRCA mutation was found in 47 (33.6%) and tBRCA mutation in 53 (37.9%) patients. Tumor testing identified an additional 9/140 (6.4%) patients with somatic BRCA mutation and negative germline testing. The comparison of germline testing with tumor testing revealed a concordance rate of 93.5% and a negative predictive value of tumor testing of 96.0%. After BRCA variants of uncertain significance were included in the analysis, concordance rate decreased to 90.9%. Conclusion: Tumor testing identified the majority of pathogenic germline BRCA mutations but missed three (2.1%) patients. In contrast, nine (6.4%) patients harboring a somatic BRCA mutation would have been missed by gBRCA testing only.

## 1. Introduction

Germline mutations in BRCA1 and BRCA2 genes occur in up to 18% of patients with high-grade serous ovarian carcinomas (HGSOC), which are responsible for the vast majority of ovarian cancer deaths [1,2]. In addition, in up to 8% of affected patients, somatic BRCA1 and BRCA2 mutations can be found in tumor specimens [3,4]. This has therapeutic implications, since patients carrying a germline or somatic BRCA1/2 mutation have been shown to respond better to platinum-based therapy and have improved overall survival [5].

In recent years, the introduction of poly (ADP-ribose) polymerase-inhibitors (PARPi) has significantly increased treatment options in cancers harboring BRCA1/2 mutations [5]. To date, olaparib, niraparib, and rucaparib have received US Food and Drug Administration (FDA) and European Medicines Agency (EMA) approval for the treatment of patients with platinum-sensitive relapsed HGSOC who are in complete or partial response to platinum-based chemotherapy [3]. The SOLO 1 trial having yielded striking results, olaparib is now used as maintenance treatment of newly diagnosed patients with advanced BRCA-mutated (germline and/or somatic) HGSOC who are in complete or partial response to platinum-based chemotherapy [6]. The evaluation of BRCA mutational status is therefore a standard procedure in determining individual adjuvant treatment decisions but also permits identification of a potential familial cancer predisposition for the patient’s relatives.

Tumor testing for the presence of BRCA mutations performed on formalin-fixed paraffin-embedded (FFPE) samples usually detects both somatic and germline mutations; however, the discrimination is sometimes compromised by technical limitations [3,7]. Patients harboring BRCA1/2 mutations in tumor specimens should therefore be referred to genetic counselling to identify a familial cancer predisposition. The detection of BRCA1/2 mutations on tumor tissue can be challenging due to several reasons: the type of possible mutations in these genes; the fact that these mutations may be found in any part of these genes, which are very large and, as such, require the entire gene to be sequenced; and the limited quantity and low quality of DNA available from routine diagnostic FFPE tissue [8]. On the contrary, germline testing of BRCA misses about 3–39% of patients with acquired BRCA mutations that are present only in cancer cells, which may have implications for treatment decisions [3,5,7,9]. Currently, there is no consensus regarding the order in which one should undertake BRCA germline and tumor testing in EOC patients, but it is generally recommended to perform both [10,11].

The aim of this study was to compare tumor BRCA1/2 status with germline mutational status to evaluate the reliability of tumor testing results.

## 2. Results 

The majority of our 140 patients had The International Federation of Gynecology and Obstetrics (FIGO) stage III or IV disease (89.4%) and high-grade serous histology (93.6%). A known family history of breast or ovarian cancer was observed in 14.2% of patients overall and in 31.0% of patients with a gBRCA1/2 mutation in particular.

Patients’ characteristics are shown in Table 1. There were no associations between BRCA status and patients’ characteristics except for family history. A positive family history was significantly associated with gBRCA1/2 mutation.

### 2.1. Pathogenic BRCA Mutations

Primary analysis was restricted to verified pathogenic or likely pathogenic BRCA1/2 variants, i.e., class 4 and 5 variants defined by the IARC unclassified genetic variants working group [12].

Of 140 patients with EOC, 47 (33.6%) patients had a pathogenic germline BRCA1/2 DNA sequence variant. Of these, 36 (25.7%) patients presented with a gBRCA1 mutation, 11 (7.8%) with a gBRCA2 mutation.

A pathogenic tumor BRCA1/2 DNA sequence variant in tumor specimens was found in 53 (37.9%) patients. In detail, 36 (25.7%) patients presented with tBRCA1 mutation and 17 (12.1%) with tBRCA2 mutation. In nine (17.0%) out of these 53 patients without a pathogenic gBRCA1/2 mutation, somatic BRCA1/2 mutation was observed in tumor specimens (Table 2, Figure 1). The types of variants of germline and tumor analysis are further described in Figure 2a,b. Three common germline mutation variants were found in multiple patients: c.181T>G p.(Cys61Gly) in three patients, c.5266dupC p.(Gln1756fs) in three patients, and c.1687C>T p.(Gln563*) in three patients.

Three patients with verified pathogenic germline BRCA variant were missed by tumor testing and a false diagnostic report generated. These three discordant cases were further analyzed. In the first case of a deletion of exon 20 in BRCA1, re-evaluation of the bioinformatic variant calls revealed the deletion also in the tumor DNA and led to a correction of the initial diagnostic report. In the second case, a germline BRCA1 frameshift mutation (c.1881_1884delCAGT p.(Ser628fs)) was not recognized by tumor sequencing because of poor coverage and sequence quality at the mutation site. Only after reevaluation, the mutation was detected by resequencing with newly extracted tumor DNA. In the third case with a BRCA2 frameshift mutation (c.8537_8538delAG p.(Glu2846fs)), only resequencing with newly extracted tumor DNA detected the germline mutation. The primary tumor sequencing four years earlier had sufficient coverage at the mutation site, but nonetheless the mutation had not been identified. Analysis of single-nucleotide polymorphism (SNPs) of both samples verified a sample mix-up.

When comparing germline testing with results from tumor testing, we found a concordance rate for pathogenic mutations of 93.6% (44/47). As was to be expected, tumor testing identified more patients with pathogenic BRCA variants (53 (37.9%) patients) than germline testing (47 (33.6%) patients). Of note, tumor testing missed three patients with verified pathogenic germline BRCA variants. Therefore, the negative predictive value of the tumor test was 96.0%.

### 2.2. Pathogenic BRCA Mutations Including BRCA Variants of Uncertain Significance (VUS)

Secondary analyses included not only class 4 and 5 variants but also class 3 variants, i.e., variants of uncertain significance (VUS). Interestingly, the concordance rate of tumor testing compared to germline testing decreased after inclusion of BRCA VUS to 90.9% (50/55) and the negative predictive value to 93.5% in contrast to 96.0% negative predictive values observed in primary analyses, where only class 4 and 5 variants had been included.

When performing germline testing in blood samples, BRCA VUS was reported in eight (5.7%) patients (two patients with gBRCA1 VUS and six patients with gBRCA2 VUS). When performing tumor testing in tumor samples, six (4.3%) patients harbored a BRCA VUS (2 patients with tBRCA1 VUS and 4 patients with tBRCA2 VUS) (Figure 3).

Discordant cases were ascertained in 22/140 (15.7%) patients. Ten out of 22 (45.5%) discordant cases were due to different VUS classifications (Table 3, Figure 4). Nine out of twenty-two (40.9%) discordant cases were due to somatic mutations detected in tumor specimens. The remaining three discordant cases were those overlooked in tumor testing with BRCA germline mutations as described further above.

## 3. Discussion

Due to the introduction of PARPi in first-line treatment of high-grade epithelial ovarian cancer, reliable tests to identify patients eligible for this treatment are crucial. Moreover, it is essential to identify patients with BRCA germline mutations in order to counsel and identify patients and their relatives with an elevated risk for breast or ovarian cancer and prostate cancer in male relatives. The present study evaluated the reliability of tumor DNA sequencing compared to germline testing in blood samples. When restricting analysis to BRCA class 4 and 5 variants, the concordance rate for tumor testing compared to germline testing was 93.5%, and the negative predictive value was 96.0%, i.e., three patients with pathogenic gBRCA1/2 mutation were missed by tumor testing. When including BRCA VUS, i.e., class 3 variants, the concordance rate for tumor testing compared to germline testing was 90.9%, and the negative predictive value was 93.5%.

This is in line with current guidelines recommending tumor testing as the primary triage test to identify the maximum number of patients eligible for PARPi treatment. Moreover, tumor testing for BRCA1/2 mutations is a cost-effective method of triaging women with EOC for genetic counseling and a confirmatory germline test to identify BRCA1/2 mutation carriers [13]. The technical ability to identify all variants through tumor testing is of utmost importance when considering the potential of a tumor-first testing model. In recent literature, it has been demonstrated that up to 5% of germline variants will be missed with tumor analysis [11,14]. In an Italian prospective study comparing BRCA1/2 germline and tumor testing results from 62 patients with EOC, the concordance between tumor and germline BRCA tests was 87.1% (54 of 62), and the negative predictive value of the tumor test was 100% [15]. During the SOLO-1 trial, out of 341 patients with pathogenic germline BRCA mutations, 12 had a tumor BRCA wildtype result, and 5 were classified as having BRCA VUS. The false-negative rate in this study was 5% [6]. The discordances between germline and tumor BRCA results were explained by differences in test coverage, variant classification, and detection of large rearrangements. In a Canadian study performing tumor and germline testing for BRCA mutations in 200 patients with HGSOC, a 100% detection rate of germline variants through tumor testing was demonstrated [16].

Of note, tumor tissue testing in our center missed three patients with verified pathogenic germline BRCA variant when analyzing the cases for the first time as part of routine work-up. In one case with a deletion of exon 20 in BRCA1, re-evaluation of the bioinformatic variant calls revealed the deletion also in the tumor DNA and led to a correction of the initial diagnostic report. In the second sample, a BRCA1 frameshift mutation (c.2806_2809delAAAC p.(Ala938fs)) was not recognized by tumor sequencing because of poor coverage and sequence quality at the mutation site. The mutation was detected in a resequencing run three years later with newly extracted tumor DNA. In the third case with a BRCA2 frameshift mutation (c.8537_8538delAG p.(Glu2846fs)) only resequencing with newly extracted tumor DNA detected the germline mutation. The primary tumor sequencing four years earlier had sufficient coverage at the mutation site, but nonetheless, the mutation had not been identified. To clarify if this discrepancy might be due to a sample mix-up, we looked to the single-nucleotide polymorphism (SNPs) of both samples. SNP analysis had proven that there must have been a sample mix-up, as 20/24 SNPs were discordant between the two samples.

As a second step, we did not restrict analyses to class 4 and 5 variants only but also included class 3 variants. Within these analyses, concordance rate and negative predictive value dropped to 90.9% and 93.5%, respectively. In our study, a significant number of discrepant cases (10/140) were due to the Department of General Gynecology and Gynecologic Oncology and the Department of Pathology coming up with different interpretations of VUS. VUS are alterations in the DNA sequence of a gene that have an unknown effect on the function of the gene product or on the risk of disease [17]. In an analysis of BRCA1/2 sequences from 10,000 individuals, 13% were observed to harbor a BRCA1/2 VUS [18]. In our population, we found eight (5.7%) gBRCA1/2 VUS and six (4.3%) with tumor BRCA1/2 VUS. The management of these uncertain results remains controversial. Specific guidelines are necessary for the correct handling of these results by both the gynecologic oncologist for therapeutic purposes and the clinical geneticist for genetic risk assessment.

Discrepancies between germline and tumor testing for BRCA mutations arise from several causes. The potential limitations of the tumor testing include the issue of accurate identification of large rearrangements as well as the tumor heterogeneity associated with the neoplastic growth [19,20]. Furthermore, computational pipelines designed to identify somatic mutations may not recognize some damaging germline variants. Mutation detection threshold levels will need to be modified to detect potentially low-variant frequencies of somatic mutations and germline mutations. There is no set recommendation as to what this level should be, as this depends on the performance criteria for the test, including any pre-analytical steps [10].

In several studies, cases showing BRCA1/2 negative status in the primary tumor and a subsequent BRCA1/2 positive relapse are described [21,22]. In our study cohort, we used tumor specimens of primary tumors in 84.4% and recurrent disease in 15.6% for BRCA tumor testing but could not observe differences in test accuracy between primary tumor and recurrent disease. Although the tumor testing of metastatic specimens at the time of progression may provide a more accurate indication of tumors likely to respond to PARPi treatment, the current literature relates to the analysis of primary ovarian tumors [10].

There is still controversy if hereditary BRCA1/2 mutation-related tumors represent a separate phenotypic identity. Since hereditary ovarian cancer is linked to hereditary breast cancer, we analyzed the number of different variants in our cohort and tried to find an association with breast cancer subtypes. In our population, 16 patients had a history of breast cancer, of which 14 had a gBRCA1, and two a gBRCA2 mutation. In 11/14 patients with gBRCA1 mutation, tumor histology was available. Six patients had a TNBC, four luminal A tumors, and one patient a luminal B tumor. The most commonly occurring mutation in breast cancer patients in our cohort is BRCA1-1687C>T p.(Gln563*), which we found in three individuals. We found no common variants for BRCA2. BRCA1-related breast cancer is often associated with triple negative breast cancer (TNBC) subtype, whereas BRCA2-associated tumors tend to be luminal-like breast cancer. In a study analyzing 531 breast cancer patients, BRCA1-633delC was detected with relatively higher prevalence in patients with TNBC, whereas BRCA2-1466delT was found mainly in luminal B tumors [23]. In contrast, a recent study found that BRCA1/2 germline-related breast and ovarian cancers did not represent a unique phenotypic identity, but they express a range of phenotypes similar to sporadic cancers [24].

A significant heterogeneity in the prevalence of germline pathogenic variants in BRCA1/2 genes has been demonstrated in different populations. The most frequent mutations in Ashkenazi Jew and non-Ashkenazi Jew origins were 185delAG and 6174delT. In Non-Jewish Caucasians, the widest variation of >20 mutation subtypes was observed [25]. In 54 families with breast/ovarian cancer, three truncation mutations in the BRCA1 gene and five in the BRCA2 gene were found. Three of these mutations have not been previously described: 308insA in one family and 3936 C>T in two families for BRCA1 and 4970insTG in one family for BRCA2. And again, in two families having Ashkenazi Jewish ancestors, the mutations 185delAG and 6174delT were found [26]. In a Sicilian cohort unlike other Italian and European regions, thirty pathogenic variants were more frequently observed, of which only some of these showed a specific territorial prevalence [27]. All our patients were of Caucasian origin. We ascertained 47 germline BRCA mutations in total, of which we found only three repetitive mutations each in three patients. Two of these mutations, i.e., 3x c.181T>G p.(Cys61Gly) and 3x c.1687C>T p.(Gln563 *), have been reported to be common in the Caucasian population [28,29]. One mutation, i.e., c.5266dupC p.(Gln1756fs) has been described to be common in the Ashkenazi Jewish population.

This study has several potential limitations. It is based on a limited cohort of patients who were treated at a single academic center, which limits the precision and generalizability of the reported mutation rates. We did not investigate the presence of alterations of other genes of the homologous recombination system, although germline variants in other ovarian cancer risk genes have been identified in 4–7% of ovarian cancer patients [30,31].

A strength of our comparative study is the independent analysis of germline and tumor mutations in two separate laboratories employing different sequencing gene panels, sequencing platforms, and bioinformatic tools. Moreover, we present real-life data reflecting clinical practice with all the pitfalls that may occur in routine testing.

In conclusion, the results of our study demonstrate that large-scale tumor testing is both effective and feasible after a certain learning curve for pathologists. In 6.4% of patients, BRCA tumor testing could detect spontaneous sBRCA1/2 mutations in tumor specimens, which would otherwise have been missed by BRCA germline testing only. We found discordant test results between tumor and germline testing in 9.3%, but most of these discordant cases were due to different VUS classification, which would not be relevant for PARP-inhibitor initiation.

## 4. Materials and Methods

### 4.1. Study Population

The present analysis includes 140 patients who were diagnosed with and treated for EOC between 2003 and 2019 at the Medical University of Vienna (Comprehensive Cancer Center), Austria. All of these patients underwent germline testing for BRCA mutations at the Department of General Gynecology and Gynecologic Oncology, Medical University of Vienna, and tumor testing at the Department of Pathology, Medical University of Vienna. Due to hospital policy, we referred all patients with primary epithelial ovarian cancer to genetic testing as well as tumor testing since January 2019. Moreover, patients with platinum-sensitive recurrent epithelial ovarian cancer potentially eligible for PARP-inhibitor therapy were referred to genetic testing as well as tumor testing. All patients available with test results for both tests were included in the present study.

The clinical data were retrieved from medical records. The histological subtype and grading were determined by gynecologic pathologists. The family history was considered positive when at least one first-degree relative had a BRCA1/2-related cancer diagnosis. All patients were treated in accordance with the standards of our institution with upfront surgery and adjuvant platinum-based chemotherapy or neoadjuvant chemotherapy and intervention debulking surgery. Surgical staging according to FIGO guidelines was performed, including hysterectomy, bilateral salpingo-oophorectomy, pelvic and/or para-aortic lymphadenectomy, appendectomy, omentectomy, and cytoreductive surgery to resect all gross tumour masses.

If EOC was suspected, the gynecologic oncologist would request the tumor and germline BRCA test upon discussions with the patients and providing written information on the potential test results and implications prior to surgery. The specific informed consent was obtained by the clinician and forwarded with the BRCA test request to the Department of Pathology, where the tumor BRCA test was performed, and to the Department of Obstetrics and Gynecology, where the germline BRCA analysis was performed. After the surgical excision, all specimens were sent unfixed to the pathology units, where they were fixed in formalin. In a next step, the samples were routinely processed and paraffin embedded to obtain histological sections stained with hematoxylin-eosin and by immunohistochemistry.

The patients received and discussed the test results with the gynecologic oncologist.

### 4.2. Germline Testing

If patients were suspected to carry germline BRCA1/2 mutations and fulfilled one of the clinical criteria for genetic testing [32], they were invited to attend a genetic assessment in the clinic and to provide informed consent and a blood sample for molecular analyses.

Genetic testing for BRCA mutation was conducted in the University Hospital since 1995 using denaturing high-performance liquid chromatography (dHPLC) and multiplex ligation-dependent probe amplification (MLPA). MLPA was performed to identify large detections and duplications and to further confirm mutations in the suspected gene. From 2007 to 2015, Sanger sequencing was performed in conjunction with MLPA, and from 2015 onwards, multigene panel testing was performed using the Illumina TruSight Cancer panel on the MiSeq instrument according to the manufacturer’s instructions (Illumina, San Diego, CA, USA). Data analysis was performed using Sophia DDM^®^ software (Sophia Genetics, Boston, MA, USA). Once a BRCA mutation was identified, the mutation was further classified, based on the probability of pathogenicity, for further risk assessment using ClinVar [33]. The genetic sequence variants were classified into one of the five categories, ranging from class 5 (i.e., pathogenic) to class 1 (i.e., not pathogenic) [34]. The results of the testing were disclosed to and discussed with the patient in a second counseling session in the hospital.

### 4.3. Tumor Testing

DNA was purified from paraffin-embedded tissue blocks of high-grade serous ovarian carcinomas with EZ1 DNA Tissue Kit (Qiagen, Hilden, Germany) on an EZ1 Advanced XL instrument (Qiagen). A quantity of 10 ng of DNA was utilized for the generation of next-generation sequencing libraries with the Oncomine BRCA Research Assay (Thermo Fisher Scientific, Waltham, MA, USA). The assay utilizes multiplex PCR-based amplicons and fully covers all BRCA1 and BRCA2 exons and an average of 64 bases of flanking sequence into the introns upstream and downstream of each exon. The Ion Chef instrument was employed for library and template generation and chip loading (Thermo Fisher Scientific). The libraries were sequenced with an Ion S5 system (Thermo Fisher Scientific). The sequences were analyzed with the Ion Reporter Software (Thermo Fisher Scientific). A threshold of 5% allele frequency was set for sequence variant calling. The DNA sequence reference databases BRCA Exchange, ClinVar, COSMIC, and dbSNP were used for variant classification. Variants were grouped into pathogenic, likely pathogenic, variant of unknown significance (VUS), and likely benign/benign. Only pathogenic/likely pathogenic sequence variants and VUS were included in the clinical report.

## Figures and Tables

**Figure 1 jpm-11-00593-f001:**
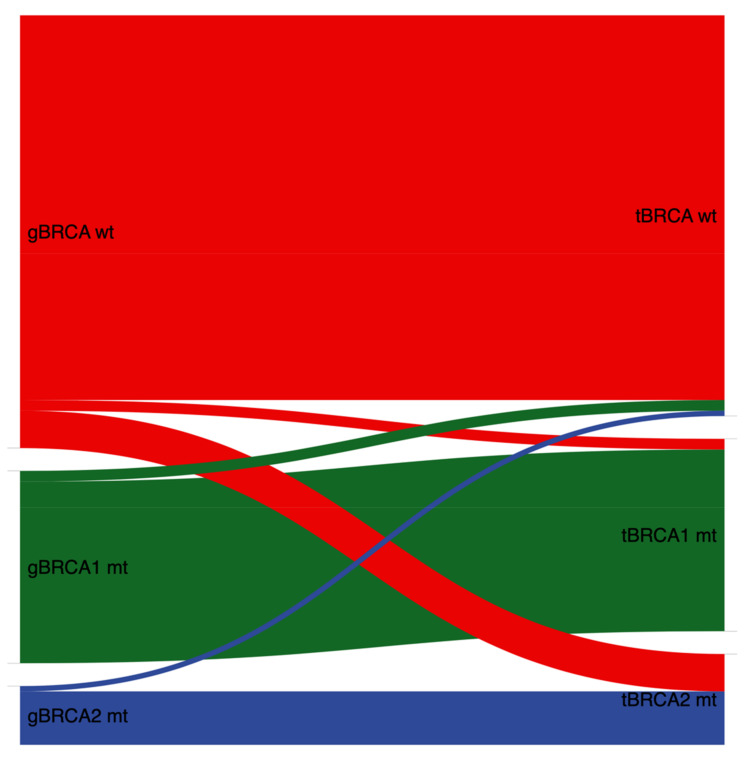
Association between tumor and germline BRCA1/2 status in 128 EOC patients (excluding BRCA1/2 VUS). gBRCA, germline BRCA; wt, wildtype; mt, mutation; tBRCA, tumor BRCA; VUS, variants of uncertain significance; EOC, epithelial ovarian cancer.

**Figure 2 jpm-11-00593-f002:**
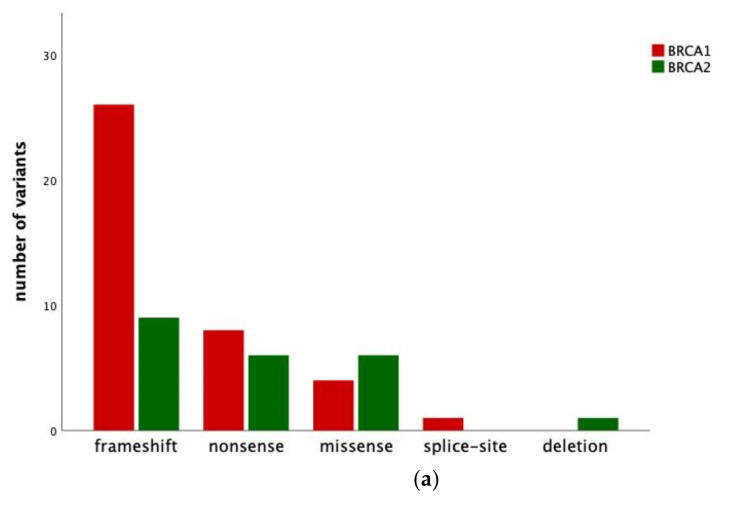
(**a**) Number and type of mutation variants detected in patients with epithelial ovarian cancer and tumor BRCA1/2 mutation. (**b**) Number and type of mutation variants detected in patients with epithelial ovarian cancer and germline BRCA1/2 mutation. n.a., not available.

**Figure 3 jpm-11-00593-f003:**
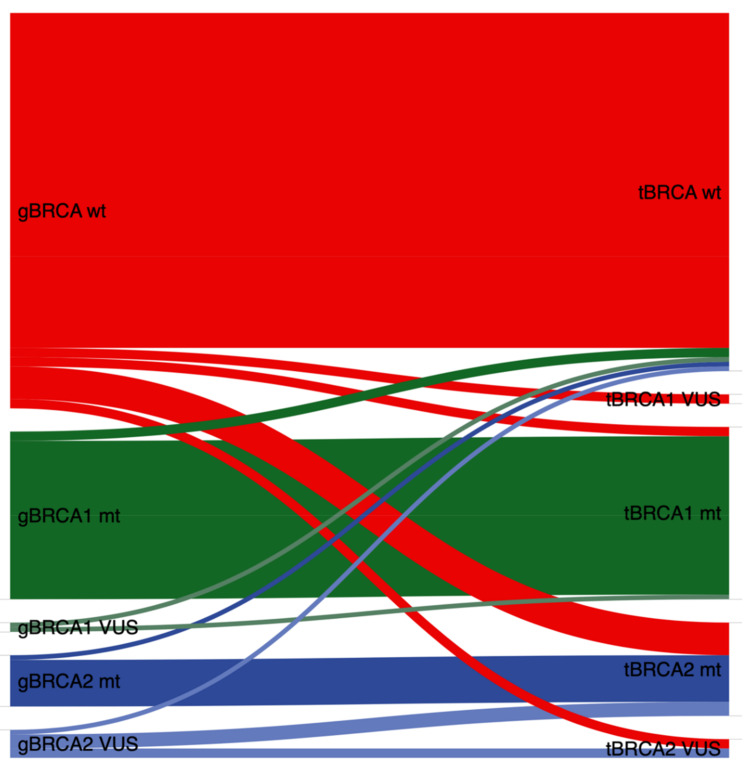
Association between tumor and germline BRCA1/2 status in 140 EOC patients (including BRCA1/2 VUS). gBRCA, germline BRCA; wt, wildtype; mt, mutation; tBRCA, tumor BRCA; VUS, variants of uncertain significance; OC, epithelial ovarian cancer.

**Figure 4 jpm-11-00593-f004:**
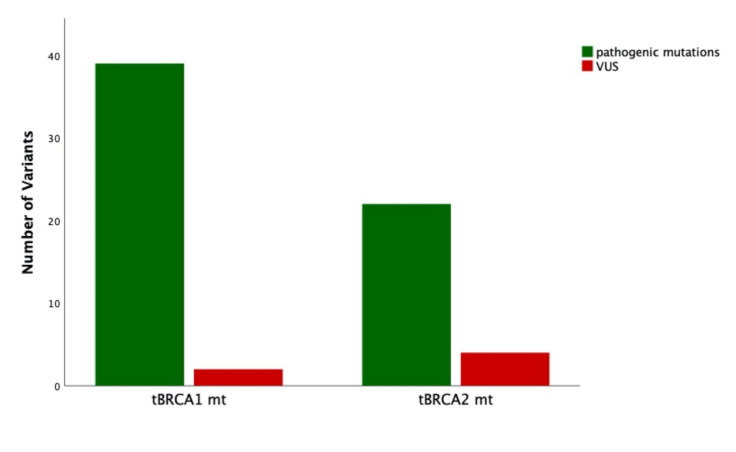
Distribution of pathogenic mutations and variants of uncertain significance broken down by BRCA1 and BRCA2 mutations in tumor specimens of 64 patients with EOC. tBRCA, tumor BRCA; VUS, variants of uncertain significance; EOC, epithelial ovarian cancer.

**Table 1 jpm-11-00593-t001:** Patients’ characteristics of the entire 140 patients with epithelial ovarian cancer, broken down by gBRCA mutation status.

	gBRCA mt (*n* = 47)	gBRCA VUS (*n* = 8)	gBRCA wt (*n* = 85)	*p*-Value
Age (years), mean (SD)	54.23 (9.67)	56.75 (11.71)	56.19 (13.34)	0.650 ^a^
Ethnicity				
Caucasians	47 (100%)	8 (100%)	85 (100%)	
FIGO Stage, number (%)				0.339 ^b^
I	2 (4.3%)	1 (12.5%)	2 (2.4%)	
II	3 (6.4%)	0	1 (1.2%)	
III	34 (72.3%)	6 (75.0%)	66 (77.6%)	
IV	8 (17.0%)	1 (12.5%)	11 (12.9%)	
missing	0	0	5 (5.9%)	
Grade, number (%)				0.255 ^b^
Low-grade	0	0	4 (4.7%)	
High-grade	47 (100%)	8 (100%)	79 (92.9%)	
Missing	0	0	2 (2.4%)	
Histology, number (%)				0.064 ^b^
Serous	47 (100%)	8 (100%)	77 (90.6%)	
others	0	0	8 (9.4%)	
Primary vs. recurrent disease for tumor testing, number (%)				0.940 ^b^
Primary disease	40 (85.1%)	7 (87.5%)	71 (83.5%)	
Recurrent disease	7 (14.9%)	1 (12.5%)	14 (16.5%)	
Family history, number (%)				<0.001 ^b^
Positive	16 (34.0%)	1 (12.5%)	3 (3.5%)	
Negative	4 (8.5%)	5 (62.5%)	30 (35.3%)	
Unknown	27 (57.4%)	2 (25.0%)	52 (61.2%)	
PFS (months), mean (SD)	30.06 (25.86)	30.13 (16.82)	32.82 (28.42)	0.843 ^a^
OS (months), mean (SD)	49.17 (44.92)	41.25 (19.85)	55.14 (44.82)	0.579 ^a^

SD, standard deviation; ^a^ one-way ANOVA; ^b^ chi-squared test.

**Table 2 jpm-11-00593-t002:** Association between tumor and germline BRCA1/2 status in 128 epithelial ovarian cancer patients (excluding BRCA1/2 VUS).

		Germline		
		Wildtype (*n* = 81)	BRCA1 mt (*n* = 36)	BRCA2 mt (*n* = 11)
Tumor	wildtype	72 (88.9%)	2 (5.6%)	1 (9.1%)
	BRCA1 mt	2 (2.4%)	34 (94.4%)	0
	BRCA2 mt	7 (8.6%)	0	10 (90.9%)

mt, mutation.

**Table 3 jpm-11-00593-t003:** Association between tumor and germline BRCA1/2 status in 140 epithelial ovarian cancer patients.

		Germline				
		Wildtype (*n* = 85)	BRCA1 mt (*n* = 36)	BRCA1 VUS (*n* = 2)	BRCA2 mt (*n* = 11)	BRCA2 VUS (*n* = 6)
Tumor	wildtype	72 (84.7%)	2 (5.6%)	1 (50.0%)	1 (9.1%)	1 (16.7%)
	BRCA1 mt	2 (2.4%)	34 (94.4%)	1 (50.0%)	0	0
	BRCA1 VUS	2 (2.4%)	0	0	0	0
	BRCA2 mt	7 (8.2%)	0	0	10 (90.9%)	3 (50.0%)
	BRCA2 VUS	2 (2.4%)	0	0	0	2 (33.3%)

mt, mutation; VUS, variants of uncertain significance.

## Data Availability

The data presented in this study are available upon request from the corresponding author. The data are not publicly available due to privacy and ethical reasons.

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
