# Peer review of "Reliability of Tumor Testing Compared to Germline Testing for Detecting BRCA1 and BRCA2 Mutations in Patients with Epithelial Ovarian Cancer"

_jpm, 2021, doi:10.3390/jpm11070593_

Round 1

Reviewer 1 Report

This is an well-written manuscript on a relatively large series of cases that assesses the concordance of separate and independent tumour and germline testing for BRCA1/2 alterations. The authors present the results clearly and succinctly and the final results and discussions are of interest for the clinical audience.

Comments: 

  • It's important the authors make a clear and consistent distinction between somatic variants (i.e. known mutation in tumour without germline variant) and tumour variants (i.e. may or may not be of germline origin). An example of wrong use of "somatic" and sBRCA is page 2, last paragraph: surely the authors found 54 tumour variants, of which I believe 9 may have been somatic. The paragraph needs redacting.
  • Table 1 and the text is confusing in a few places in the manuscript. Table 1 suggests 2 gBRCA1 variants were missed in the tumour sequencing, yet the text (page 3 last paragraph and page 4 first paragraph) mentions one BRCA1 and one BRCA2. Moreover, in the discussions the authors mention another variant (large exon 20 del) that was originally missed by tBRCA testing. This should also be included in Table 1 and in the results.
  • The authors suggest sample mix-up for the first discrepancy. If this was the case it can be very easy to assessed by looking at all the SNPs covered by the panel in both samples and see if it was the same patient or not. This aspect needs to be clarified.
  • In the first paragraph of the discussion the authors mention relatives counselling for risk of breast and ovarian cancer. This should include prostate cancer as well for male siblings, particularly when BRCA2 germline mutations are found.

Reviewer 2 Report

In the present study, the authors evaluated the reliability of tumor DNA sequencing compared with germline testing in blood samples and found a concordance rate of tumor testing compared with germline testing of 95.7% when only likely pathogenic/pathogenic BRCA variants (class 4-5) were examined. In contrast, when including BRCA VUS, the concordance rate for tumor test versus germline test was 92.7%.

The paper is readily intelligible because the experimental design is well-constructed, clear and well described with figures appropriate to the subject matter. The scientific background and aims are clearly explained. The methodology seems to be appropriate to the research field and the conclusions logically follow from the results. For all these reasons, the paper could be considered adequate for the standards required for the publication in this journal, after minor changes only.

Minor revisions:

1) What are the most frequent germline BRCA1/2 variants (pathogenic and VUS) you've found in your analysis? And which ones in the somatic line? Please report the most frequent variants in a table.

2) Please insert a table with clinical data of patients.

3) What were the inclusion criteria for genetic testing?

4) Since hereditary ovarian cancer is linked to hereditary breast cancer in the HBOC syndrome, have you found any mutations that are found to be related to specific molecular phenotypes of breast cancer? For examples, the authors should cite and discuss the following papers: PMID: 32164626; PMID: 33403015; PMID: 16261400.

5) The authors should verify if some BRCA1/2 variants are population-specific, as shown, for example, in the following papers to cite and discuss: PMID: 24131973, PMID: 32380732, etc.
